# A database of databases for Common Era paleoclimate applications

Michael N. Evans<sup>1,2</sup>, Lucie J. Lücke<sup>2</sup>, Kevin J. Fan<sup>3</sup>, and Feng Zhu<sup>4</sup>

<sup>1</sup>Department of Geology and Earth System Science Interdisciplinary Center, University of Maryland, College Park, MD, USA
 <sup>2</sup>School of Geosciences, University of Edinburgh, Edinburgh, UK
 <sup>3</sup>Computer Science and Geology, University of Maryland, College Park, MD, USA
 <sup>4</sup>NSF National Center for Atmospheric Research, Boulder, CO USA

Correspondence: Michael N. Evans (mnevans@umd.edu)

Abstract. We present a database of curated databases (DoD2k version 1) developed for Common Era (1-2000 A.D.) paleoclimate research. The DoD2k leverages existing community efforts, many of which arise from the PAGES (Past Global Changes) 2k working group, and the codebase developed by the paleoclimate data informatics communities over the past decade. Using a common, compact set of terms for metadata and data management, we merge five existing curated databases. These individual

- curated databases represent a range of approaches, from single archive-single observation to multiarchive-multiobservation collections, and span a total of 14 archives, 49 data types, and 4613 records within the Common Era. We then use a multistage algorithm to remove duplicates, checking against a common set of metadata and comparison metrics. We illustrate the value of the DoD2k with two applications. In the first, we extract the moisture and temperature subset of records and perform an empirical orthogonal function (EOF) analysis on the resulting multi-archive, multi-observation dataset. In the second, we show
- that calcite speleothem oxygen isotopic composition is consistent with proxy system simulations. DoD2k may also be useful for paleoclimatic detection and attribution analysis using proxy system modeling, data assimilation, and deep learning for the development and testing of improved proxy system models.

## 1 Introduction

The climate is changing, forced primarily by human-caused increases in greenhouse gas concentrations, aerosols and land use change, toward a warmer and more moisture-inequitable state, in which extreme events are more likely, and more extreme, than observed during the 20th century (Arias et al., 2021). Superimposed on that are other causes of climate variation and change, for instance, arising from volcanic activity, solar and orbital variations, as well as the tendency of the climate to wander without any forcing at all: the internal, unforced variability. If the former is thought to be well understood, the latter is not: how the climate system, broadly defined as the coupled ocean, atmosphere, land surface, land and sea ice, biota, and solid earth integrates and

20 responds to such natural forcings, may take tens to thousands of years to be fully realized (Miller et al., 2012; McGregor et al., 2015; Abram et al., 2016; Gebbie and Huybers, 2019). How then, to define both the spatial imprint and the amplitude of the climate change that arises from such forcings, and distinguish it from the human-driven forcings? The answers are important: first, for defining the equilibrium and transient climate change in response to a unit of forcing, over time and in different parts of the world (Forster et al., 2021); second, for detecting and projecting the impacts of both anthropogenic and natural climate

forcing over past and future decades and centuries (Fox-Kemper et al., 2021; Marvel et al., 2019b, a). For such goals we need realistically forced paleoclimate simulations and observations (Neukom et al., 2019a).

The development of the observational target for such work is the focus of the present contribution, in particular for the socalled Common Era (1-2000 CE), for which observations from paleoclimatic archives are most dense and diverse, and permit an approximate 10-fold increase in the time interval of study relative to the historical record. More specifically, we would

- desire the most dense and random sampling in space and time, of all possible observations, imprinted with a diverse set of climatic information, and resolving timescales of variation from subannual to multicentennial with similar observational temporal resolution, and with well-characterized chronological uncertainty. The natural starting point for such an effort would be public repositories of individual paleoenvironmental datasets and databases, such as at the National Center for Environmental Information (https://www.ncei.noaa.gov/products/paleoclimatology) and PANGAEA (https://pangaea.de). However, this is
- impractical for multiple reasons, including some nonuniformity in dataset submissions and metadata, changes in repository submission templates and requirements over time, and the presence of multiple versions of datasets in repositories whose prime directive is preservation and availability (Anderson et al., 2019).

An alternate foundation for the development of such a database is in existing databases, many of them compiled by yearslong efforts by PAGES (Past Global Changes; www.pastglobalchanges.org) Working Groups. PAGES databases are the result

- of leveraging community-level specialist expertise that is difficult for any single research group to assemble or maintain. The work of many individuals in multiple groups has enabled the development of publicly available observational datasets, the metadata that describes them, and most recently, the open semantic formalisms (Emile-Geay and Eshleman, 2013) and codebases (McKay and Emile-Geay, 2016) that enable their re-use. However, each such database, although rich in metadata, metadata uniformity, error checking and quality control, is generally assembled for a specific purpose. For example, the PAGES
- 2k Consortium (Consortium, 2013; Emile-Geay et al., 2017a) originally planned development of a multiarchive database (wood, coral, ice, documents, lake and marine sediments) of many different temperature-sensitive observations in these archives for the purpose of global mean and spatially resolved surface temperature reconstructions (Neukom et al., 2019a, b). The SISAL Working Group (Kaushal et al., 2024a) developed a single (speleothem) archive of multiple observations (e.g.  $\delta^{18}$ O, Mg/Ca) made in that particular archive. The Iso2k Working Group (Konecky et al., 2020a) produced a multiarchive (marine sediment,
- lake sediment, marine carbonate, speleothem carbonate, wood, ice) database of solely  $\delta^{18}$ O and  $\delta$ D observations in those archives, agnostic of climatic interpretation. For facilitating the repurposing of these databases for other scientific goals, such as the reconstruction of hydroclimatic variability (Falster et al., 2023) and the detection and attribution of climate change in both moisture and temperature via paleoclimatic data modeling (Franke et al., 2022a), we might need to combine multiple existing databases.
- There are multiple challenges to creating a unified database of databases for Common Era paleoclimate applications. These include differences in the metadata and terminology for describing datasets across databases but within even the same proxy observation types and biogeochemical archival materials; differences between databases of the required metadata, sampling resolution, age model development, time resolution, level of replication, descriptions of observational uncertainty, and interpretational notes; and the problem of duplicate detection across combined databases (Anderson et al., 2019; Tardif et al.,

2019; Steiger et al., 2022). Unfortunately, differences in metadata and terminology for defining properties across different curated databases, as well as differences in database terminology, structure, management, make merging databases and cleaning them for duplicates difficult. For instance, PAGES2K (Emile-Geay et al., 2017a) has 173 dictionary terms, and as it happens, SISALv3 (Kaushal et al., 2024a) has 173 unique dictionary terms linking its 21 constituent csv files into a database. However, these are not the same 173 dictionary terms as for PAGES2k. Although there is common overlap in metadata, such as site identification name, they have different keys ('paleoData\_TSid' in PAGES2K, 'site\_id' in SISALv3).

Here we describe an approach to resolving these challenges with a flexible and open framework, implemented as Python functions, scripts and Jupyter notebooks, in which a standard set of metadata by which to merge existing databases could be specified (McKay and Emile-Geay, 2016), and in which duplicates across merged datasets could be identified and removed. The framework is extensible and can incorporate new databases or updates to existing ones. It provides methodical and com-

70 prehensive testing for duplicate records and can be easily attached to paleoclimatic analysis and reconstruction toolsets (Zhu et al., 2023, 2024b).

The existing Common Era paleoclimate datasets to be merged are briefly described in section 2.1. The data, approach and the code base for merging are described in section 2.2. The resulting Database of Common Era paleoclimate Databases (hereinafter, DoD2k) is described in section 3, and some applications are illustrated in section 4. Conclusions and an outlook for future development are in section 5.

2 Data and Codebase

#### 2.1 Data

As a target, we assemble five such databases of Common Era paleoclimate records.

- Breitenmoser et al. (2014a); Franke et al. (2022b) (hereinafter, fe23): restandardization of tree-ring width (TRW) chronologies for comparison with their 20th century simulation using the VS-Lite data model (Tolwinski-Ward et al., 2011), with data available within the interval 850-2000 CE and with climatic interpretations re-estimated as published in a subsequent study (Franke et al., 2022a);
  - Emile-Geay et al. (2017a, b): PAGES2k (hereinafter, p2k): multiproxy, multiarchive compilation of temperature-sensitive proxies for the Common Era, updated for records from Palmyra Atoll (Dee et al., 2020);
- Konecky et al. (2020a); Konecky and McKay (2020): Iso2k (hereinafter, iso2k): multiarchive compilation of oxygen and deuterium isotopic records extending through the Common Era and into previous periods;
  - Walter et al. (2020, 2022): CoralHydro2k (hereinafter, ch2k): single archive, multiproxy compilation of records from coral carbonates, within the Common Era;
  - Kaushal et al. (2024a, c): SISAL2k (hereinafter, sisal): single archive, multiproxy compilation of cave carbonate records, extending from the present through the Common Era and into previous periods.

#### 2.2 Workflow and Codebase

An overview of the workflow for creating DoD2k is shown in Figure 1. DoD2k as well as the codebase for its production are supplied as a GitHub repository (https://www.github.com/lluecke/dod2k). The codebase is written in Python 3, and includes a series of jupyter notebooks and Python scripts, which read in the original databases, concatenate to a large database, perform a duplicate search and eliminate duplicates, output the product (DoD2k), and provide some summary plots (Figure 1). Thus, we here present not only DoD2k, but also the opportunity for users to modify the source code based on their specific requirements, including but not limited to, add data, add variables, and make their own expert decisions regarding the potential duplicate candidates. In the following paragraphs, we present an overview of the DoD2k database and the associated Python utilities.

**Figure 1.** Schematic overview of the DoD2k workflow. From upper left: starting with the databases to be aggregated, we load each dataset as a compact common subset of the metadata and data using a standard set of dictionary terms, if necessary translating from the original terms. We then concatenate the data, perform basic metadata checks, and check for duplicates (optionally with operator review and temporal compositing) before creating the DoD2k. All operator choices are journaled and the notebooks may be commented for subsequent review and for reproducibility of processing and traceability back to original databases and their entries (Bush et al., 2020).

The codebase consists of two key parts. The first includes loading the original databases and their concatenation to a common database (referred to as the 'load notebooks'). The second involves the duplicate detection process. In addition, we supply several notebooks for data visualization purposes, and two notebooks for applications described in section 4. An overview of the notebooks and scripts associated with the key parts of the DoD2k development process (Fig. 1) is given in Table 2.

#### 2.3 Environment

A virtual environment for running the Python functions, scripts and Jupyter notebooks built within a Jupyterhub installation (https://tljh.jupyter.org/en/) can be found at the aforementioned github repository in the file cfr-env.yml. To reproduce the environment, users just need to type the following command: conda env create -f cfr-env.yml -n cfr-env.

#### 2.4 Metadata fields

To assemble a database of databases, we identified a set of 16 metadata fields (Table 1) which satisfy the following criteria across all the individual databases. Apart from a few fields specific to DoD2k, the nomenclature of the fields was largely adapted

- from Emile-Geay et al. (2017a) and the PAGES2k v2.0.0 temperature-sensitive database. Criteria were that the metadata is commonly used within the community, and the majority of the original databases have non-missing entries within the field. For a description of how the individual metadata parameters were collected, we refer back to the original databases and their development by teams of specialist researchers. Note, however, that this set of fields is by no means exhaustive, and if desired by the community or by an individual user, may be expanded. For example, the p2k database has 173 metadata fields.
- However, half of these fields are missing for 85% of entries, making it more difficult for users to identify and extract relevant metadata. We have tried to keep the majority of fields populated, and only climateInterpretation\_variableDetails and duplicateDetails have a significant number of missing entries in DoD2k. However, both these fields are complimentary with the only purpose to provide extra information where needed.

#### 2.5 Load notebooks

- The load notebooks read the five original databases (ch2k, fe23, iso2k, p2k and sisal), extract a number of shared variables, and concatenate the databases. Each notebook follows the structure:
  - set up environment, load source data
  - potentially process data according to database provider (may use code supplied by authors of the original database) to obtain a pandas dataframe
- identify and extract the relevant variables
  - convert metadata and data to the correct format
  - save output as a "compact" pandas dataframe with standardized metadata

The output of each load notebook is a standardized "compact" set of 18 metadata and data fields (Table 1). Output is saved in pickle (.pkl) format as well as a series of comma separated value files. A unique datasetId is pulled from the identifier used in the component curated dataset. Each additional metadata field contains either information from the original dataset as

published, or has been added for the purpose of improving data identification. In the case of the former, some information has
