# Peer review of "A database of databases for Common Era paleoclimate applications"

_Earth System Science Data, 2025_

## Author Comment (AC2)

**Authors' Response:**

Thank you to all three reviewers for the insightful remarks and constructive criticisms of the submitted manuscript, database, and codebase. We are pleased to describe plans for revising the manuscript as follows (reviewer remarks in blue; our response in black.)

RC1: 'Comment on essd-2025-364', Anonymous Referee #1, 20 Aug 2025

Citation: https://doi.org/10.5194/essd-2025-364-RC1

General comments (overall quality of the preprint)

**Summary:**

This manuscript presents a "database of PAGES 2k databases," or dod2k, which provides a set of tools for translating between, and combining records from, PAGES 2k databases. This type of toolkit has long been needed and I'm grateful that the author team has taken it on. Overall, the manuscript and the set of Python functions and codes are very well thought out and of high quality. They present notebooks for the applications that they showcase in the manuscript, which are useful and straightforward, and the notebooks can be run while reading the manuscript side-by-side.

I did have some concerns regarding the clarity of the workflow as presented in the manuscript and the leap that users must make between reading the paper and actually using the code, which is not as straightforward as it could be. I have made a few suggestions below for clarifying the text and the workflow figure, and for providing a more thorough Quick Start guide and a simple tutorial (which can be as simple as an example set of more thoroughly commented notebooks walking through the creation of the dod2k\_dupfree\_dupfree dataset that the authors use for their examples in the manuscript). Critically, I believe the author team should do some beta testing of these tutorials and the QuickStart guide with a few new users who have had \*no\* involvement in the project whatsoever. The errors and troubleshooting that I encountered while reviewing most likely happened because when putting together the Github repo, readme, and quick start guides, the author team was already so familiar with the code and the conceptual workflow that it was easy to take certain steps for granted. These additions will make the database/toolkit (is it a database or a toolkit?...see below) more widely usable and improve the overall accessibility of the manuscript. My comments below mostly focus on making sure a reader of the manuscript can easily understand and follow along with the process of loading the databases and removing duplicates, as well as following the example applications given.

Authors' response: We agree with the suggestion to create a more thorough Quickstart. We will implement this using MkDocs (https://www.mkdocs.org; https://github.com/mkdocs/), with a set of jupyter notebooks in tutorial style with additional commenting. We will beta-test this tutorial with new users including members of the hydroclimate2k community, as well as students wishing to filter dod2k for other applications.

**Is dod2k more accurately called a toolset than a database?**

As catchy as 'dod2k' is, I go back and forth on whether the product here is actually a database. Actually I think it is more accurately called a set of tools. Yes the result of using the tools is to load all the databases' record into one super-database, but dod2k does not have its own datasets and metadata. For example if a new temperature record is published, it would be submitted to the pages2k temperature database and added to the next release, rather than it being added to dod2k directly. In addition, the final "dod2k" product ultimately depends on decisions the user makes about duplicates—it is not the same set of records every time. So I think a more accurate title would be something like DT2k (databaseTools2k) or DIOP2k (Database InterOPerability 2k) or something similar. This is sort of semantic but the distinction is important when thinking through the process for updating the databases themselves plus the tools included in dod2k, see below.

Authors' response: Thank you for pointing this out and sorry for the confusion. We realize that we should have advertised the database property of this effort more in the manuscript as it is our major motivation. Following the aims and scope of the journal, we prefer to chart a middle course and treat this as a "database of databases" contribution.

To be clear, the database could be used as a standalone product, without running the notebooks, if the user agrees with the choices made to create it. They have the decision log that documents the choices made during operator supervision.

The purpose of creating dod2k was always to create a database of existing curated databases; this is described in Section 1. As updates are made to the underlying databases by their curators, or additional databases become available, these can be integrated by revising the individual and load\_dod2k notebooks.

On the other hand, if the user wishes to use the load and duplicate detection notebooks to make other choices or to merge other databases, they could do so. Therefore, we will also provide the essential toolkit, for which we can adopt the name dt2k as suggested. These are the load notebooks which specific to each component database; duplicate\_detection; duplicate\_decision; duplicate\_removal.

The remaining application notebooks are specific to the paper and can be used as the basis for the tutorial. We will revise the paper, sections: 2.1 Data and 2.2 Codebase and workflow to make these pathways clear.

Because new records may be added to each sub-database, we would propose to release updated dod2k versions annually for the community as our time and support permits.

Stewardship and updates to dod2k and the underlying databases:

The notebooks provided here are tailored to the most current versions of the Pages 2k

databases and include some workarounds for the bugs or peculiarities that are specific to those databases and their specific versions (for example the Palmyra record in the pages2k load notebook). That means that if/when the individual databases are updated, the dod2k code may break or some sections of the notebooks may become moot, while other sections may need to be added for the new inevitable peculiarities that arise. Given that updating the databases themselves is always a large endeavor, what is the plan for updating dod2k/dt2k/diop2k when new versions of databases are released to the community? Will updates to dod2k be governed by the author team of this paper, or will community users contribute code? If the latter, who will review and commit those changes? Can this discussion be added to the text?

**Authors' response:**

We feel that having the key jupyter notebooks on github enables branches and versions to adapt as the underlying databases are updated, or as other curated databases are added for user specific purposes. This author team could review and commit those changes, but we also think that the notebooks could be adopted by data informatics groups. We would note that because the process involves operator decisions about retention and compositing of data, users might well make different choices and produce their own versions.

For the dod2k database, we intend to request it be hosted on lipdverse.org to ensure we can make point releases that draw from updated or additional component databases relevant to Common Era paleoclimate research there that are updated in advance of their public repository counterparts (e.g. PAGES2k v2.2.0 on lipdverse vs PAGES2k v2.0.0 on Figshare and PAGES2k v2.0.0 on NCEI/Paleoclimatology.)

We will note these two points in revised Section 5 (was: Section 4.3, Outlook; see response to Julien Emile-Geay's comments below).

**Usability of the dod2k tools and some additional resources for new users:**

As is always the case with Python, the trickiest part especially for non-expert users is getting the environment set up and getting example scripts or notebooks to run without breaking. The paleoclimate community uses a variety of programming languages and it is likely that many (if not most) of the prospective users of dod2k have only a basic knowledge of Python and some (many?) users will not have any knowledge at all (compared with, for example, R, which seems to be more popular in the community). When I initially started to review the paper, I cloned the github repository and tried setting up the environment and running the notebooks, but then I started running into enough errors requiring troubleshooting that I did not feel the manuscript was ready to review. The editor reached out to the authors and they then posted a comment providing an updated .yml file for a python environment (dod2k-env.yml) and a QuickStart, both of which were very useful. I was able to successfully get most of the notebooks running. Still, there is an underlying issue which is that it feels that the dod2k workflow and tools need to be better clarified in the text, and probably more rigorously beta tested with users from the broader

paleoclimate community before it is really ready for wide release. See my comments on section 2.2 and the quickstart guide below.

Something that would help immensely for all users (regardless of their python experience) would be adding some resources to allow users to get started with the dod2k tools. At minimum, it would be helpful to include with this publication a "tutorial" that is simply a set of notebooks that exactly reproduce the steps taken in this manuscript to produce the "dod2k\_dupfree\_dupfree" version of the dod2k dataset, which appears to be the one used for the examples in sections 3 and 4. These notebooks should include more extensive commenting to walk a user through the decisions that were made to produce that version of the dataset, including in the notebook directly some comments on the rationale/justification for each of the decisions about duplicate cases. I think this is what was intended with the README file in the duplicate-screened database that is created with the screening notebooks, but the examples in the github repo just have a simple README that lacks operator comments on the duplicate screening.

For inspiration regarding tutorials, I suggest looking to the example of Pyleoclim, which provides a number of extremely accessible tutorials that are easy to run out of the box: <a href="https://github.com/LinkedEarth/PyleoTutorials">https://github.com/LinkedEarth/PyleoTutorials</a>

Authors' response: We agree and will add a tutorial that exactly reproduces the steps to create the duplicate free dod2k dataset. We will add additional commenting to the key notebooks for database loading and concatenation; then duplicate detection, decision and removal; then filtering for subsets from the dictionary terms. The final product will be renamed to clearly indicate what it is.

After doing this, a short section should then be inserted into the beginning of the Results section of the manuscript that describes the process of creating dod2k\_dupfree\_dupfree, with the names of the notebooks and the order in which they are used very clearly stated so that a reader can follow along and create their own version of dod2k\_dupfree\_dupfree if they like. This addition will also help with overall reproducibility of the figures in the manuscript.

**Authors' response:**

We note that Section 2, in particular Sections 2.5 and 2.6, including Table 2, does exactly this. We will standardize the notebook names such that the generic names listed in Table 2 map easily to the specific versions of these notebooks including in dt2k.

Specific comments (individual scientific questions/issues)

.yml file:

The original cfr-env.yml fails to complete a working environment because of a problem with pyvsl. The new yml file, dod2k-env.yml, successfully produces a working conda environment which is great. However the new .yml file did not include any version of jupyter notebook so that

had to be installed separately. I recommend adding the jupyter and notebook lines back to the dod2k-env.yml that was previously in the older .yml file.

Authors' response: jupyter and notebook are added into the dod2k-env.yml environment.

**Comments on the process of removing duplicates:**

The notebooks for identifying and handling duplicates take a very long time to run (at least 2 hours on my machine), and due to connection timeouts I was not able to successfully run the dup\_detection notebook even with multiple attempts. This is even with loading only two of the databases. Aside from connection timeouts I also encountered this error: "File Save Error for dup\_detection.ipynb. Failed to fetch." It's fine if code takes a long time to run, that's just reality. But perhaps in the manuscript and in the QuickStart/readme files, it would be helpful to specify at the outset of the section on duplicate handling that there is a reasonable version of the dod2k dataset, dod2k\_dupfree\_dupfree, that is ready to be used should the user want to do so rather than starting out with making their own decisions regarding duplicates. Many users may want to skip the full duplicate detection/decision-making process at first and just jump in to exploring a version of the compiled dataset where some reasonable decisions have already been made (eg, dod2k\_dupfree\_dupfree). This will help people get started working with all the compiled datasets right away.

I did really appreciate that the csv file already exists in the Github so that a user can just proceed without running the full duplicate detection process. That was a nice touch. It may be helpful to clarify the name of that csv file in that cell of the notebook where it directs a user to comment out that cell if they wish to use an existing csv file.

**Authors' response:**

We are sorry to hear the code took that long to run. To be clear, the user does not need to run the entire workflow. They may instead use the dod2k product, with reference to the notes on screening provided. Provision of the outputs is a workaround for those who want to see the results and build applications from these, rather than make their own duplicate decisions. The duplicate decision operator choices are logged, and we will make this more apparent in the planned tutorial and in the manuscript text, section 2.6 and Table 2.

Users may perform their own screening, starting from the detected candidate duplicates, which are provided. We will note in the text, Section 2.6 that the duplicate detection notebook takes the most time, but does not need to be rerun if the same parameters into it are specified as described in the text, because the output csv files are provided.

For those who wish to execute the entire workflow, we will investigate optimizing the code to make this step faster; in preliminary tests, small changes can decrease the automated steps by a factor of two, and reduce the number of records flagged for operator screening by about 15%, and we will test whether the screening then produces the same results.

In addition to the above, to promote useability I wonder if it is possible to change the way that duplicates are written to the csv file while it's running, so that it's possible to pick up where it left off if the notebook finishes only partway through? I get that this may not be possible, but it's worth looking in to.

**Authors' response:**

This functionality is part of the original submission. We have checked and indeed it is possible to pick up the duplicate decision operator screening process from a partially completed session. This is now described in the tutorial, and will revise the decision notebook for further transparency and usability.

Some suggestions for the decision-making for duplicates: The process of prompting the user to manually make decisions is nicely thought out here and I appreciate the decision-making metadata for future reproducibility purposes. I find the figures a little tough to parse, though. Can both Y-axes for both datasets be plotted, and then specify which Y axis pertains to which record? Also, I could not quite tell what the grey line is exactly? I see that it is something about the differences between the records but it jumps around so wildly that it was sort of hard to tell, and differences were often very far from zero even with records that had high degrees of correlation. Plus it is a little distracting and makes it hard to look very closely at the underlying records. Could it be added to a sub-panel? And/or just explained better in terms of what the units are.

**Authors' response:**

We will simplify the plotting to remove these results that are not useful for visually evaluating duplicates.

Finally, there seem to be a number of "dod2k\_dupfree" folders with different initials appended to them (and one with "\_dupfree" again appended), but no clear description of what the differences are (though I gather from the paper these are probably subsets of moisture and temperature records, and not someone's initials…?). Can this go in a readme file somewhere?

**Authors' response:**

Yes, this can be cleaned up and filenames described in a readme. The result labeled \_dupfree\_dupfree was the test to check that a second pass through duplicate detection in fact found no duplicates.

**Other specific comments on the manuscript and notebooks:**

- In line 69, the text says "The framework is extensible and can incorporate new databases or updates to existing ones." From what I can tell though, a user is supposed to use the versions of the databases that are stored in the github repository. That means the user should essentially

download each of the databases from this github repository, rather than download them from their official public repositories or websites? Which ultimately means that as updates to the individual databases are released, this github repository has to then also be updated, yes? Can this be clarified? And, this is where it would be very helpful to include an overall discussion of plans to keep dod2k updated (see my "overall" comment above).

**Authors' response:**

As a database of community curated databases, the component databases should be downloaded from their publicly accessible repositories, especially if these databases are being updated (see also response to Julien Emile-Geay's comments, below). At the same time, the github repository should probably include the particular databases we used to create the specific version of dod2k. Updates to the github repository will include updates to the source databases as we become aware of them and they become available, such as described in response to Julien Emile-Geay's comments, below. We will clarify this in the text, Section 2.5, and in the load notebooks. These all list the source url (for instance, in load\_ch2k.ipynb, cell 1), and the shell commands to obtain the dataset (for instance, in load\_ch2k.ipynb, cell5), but we will check that the database sources is standardized across all 5 load notebooks.

- Section 2.2: The workflow figure is great, but I found the actual description of the workflow in the text to be lacking some key, concrete information. For example, to load all of the databases, does one need to go through and execute each individual load notebook? Can you specify where those load notebooks are located (they aren't in the main directory of the github repo, and I found them, but it would be easy to just specify here)? Can someone load just a few of the databases, or will that break things later? I started by just loading 2 of the databases, since the duplicate detection process was so lengthy I was hoping this would cut down on time. It wasn't actually successful (see above) but if it had been, would that have worked, or does a user actually need to load in all the databases? I looked for concrete workflow examples in Section 2.5 and 2.6, and in the example applications in 3.2, but those still don't list the complete workflow to reproduce the figures made here. In addition to the tutorial I mentioned above, can you also include more specifics in Fig 1, rather than just a conceptual overview? Or another figure that lists the specific workflow for one (or both) of the example applications, including the names of the notebooks? And finally, as I said under "overall comments" include a set of notebooks that walks through the creation of dod2k\_dupfree\_dupfree including the rationale for decisions about duplicates.

**Authors' response:**

We will list the notebook load\_dod2k.ipynb in Table 2. This is the notebook that takes the compact dataframes that were created by the load notebooks and assembles them into the unified dod2k database of databases. That aggregate dataframe is the basis for duplicate detection and removal. We apologize for omitting this critical link in the workflow from Table 2, and it will be made clear in the tutorial. Modifying this notebook should allow the user to create

a dod2k from a smaller subset of the five databases we used, or to add additional databases, for instance the Burgdorf et al (2024) database DOCU-CLIM, as mentioned in the Discussion. We will note this flexibility in Section 2 when introducing and commenting on load\_dod2k.ipynb, and then in the Discussion.

We can certainly add more detail to section 2.2 to indicate where to find the notebooks listed in Table 2.

We will modify Fig 1 to include the names of notebooks as listed in Table 2.

We will create a tutorial notebook as previously suggested, with more complete documentation to clarify the workflow.

We will also add notes to Section 2.5 about the choices we made during the duplicate decision process to create the resulting dod2k database. This is memorialized in the duplicate decision output, which is called duplicate\_decisions\_INT\_YYYYMMDD.csv (INT=initials entered by operator), which is saved into the duplicate\_detection directory.

In addition to the text in section 2.2: The Quick Start guide that was added is helpful, but it should be revised to contain files in the order that one should use them, following the steps in the manuscript and in the workflow figure.

**Authors' response:**

We believe the suggested and planned tutorial is a good way to merge our response to this comment and the comment below suggesting a tutorial (see below). Following prior Author Response notes, we will also revise and expand the subsections in Section 2 to mirror the workflow described in Fig 1.

- Section 2.3: The way this is described here is not really accurate. The text says "A virtual environment for running the Python functions, scripts and Jupyter notebooks built within a Jupyterhub installation 105 (https://tljh.jupyter.org/en/) can be found at the aforementioned github repository in the file cfr-env.yml." Looking at the Github repository, it does not seem like Jupyterhub is actually necessary. I believe this text should actually say "Users can create a virtual environment to run the Python functions, scripts, and Jupyter notebooks using the dod2k-env.yml file found at the aforementioned github."

**Authors' response:**

Correct. We have added jupyter and notebook to the dod2k-env.yml and will add the reviewer's language to Section 2.2.

Unless there is some reason that users need to be using Jupyterhub? From what I can tell, users still need to download all the datasets and code in the github repository, right? If that's not the case, that needs to be clarified here as well as in the README file on github.

Anyway, this language should be clarified and ideally, more depth given for how to get this set up on an individual's laptop and a group server (probably the two most common approaches) so that it is easier for users to access dod2k and the notebook examples here. It is OK to assume some working knowledge of Python, but since the paleoclimate community will be approaching this database with very different experience levels, providing some concrete info on getting everything set up in a straightforward manner will go a very long way with the community. Clarify that all a user needs to reproduce what is done in the paper and then perform their own analyses, is the ability to use Jupyter notebooks, and to set up an environment with specific package versions etc.

For what it's worth, I did not use JupyterHub, I just cloned the github repository to an Ubuntu server that I use, and worked through all the examples there. If people need instructions on how to use Jupyter lab or Jupyter notebooks there are some helpful tutorials by LinkedEarth that you can recommend in the manuscript and/or on the Github: https://github.com/LinkedEarth/PyleoTutorials

- One thing you may wish to consider, following Pyleoclim's footsteps, is to encourage first-timer users run the notebooks with myBinder. This way they do not need to install anything. I tried to use myBinder to run the dod2k notebooks in order to review this preprint in the first place, but ran into errors importing packages when trying to run the load scripts, so something is not right there. This would be very useful for the DoD2k user community and allows people to get started checking out the codebase without having to figure things out on their local machines.

**Authors' response:**

Unfortunately we cannot use myBinder because their supported Python version is behind the version we used for dod2k. But we agree that a tutorial will be useful and will provide this in a revised submission. We are using MkDocs (<a href="https://github.com/mkdocs/mkdocs">https://github.com/mkdocs/mkdocs</a>), which creates a wiki page on github, to create this tutorial.

Technical corrections (compact listing of purely technical corrections)

**Just one minor correction:**

Line 20: "may take tens to thousands of years to be fully realized" makes it sound like it may take tens of thousands of years for people to figure out internal climate variability. I am rather more optimistic than that. I suggest rephrasing.

**Authors' response:**

We will revise this overlong and unwieldy sentence and the one before it to:

"Superimposed on that are other causes of forced climate variation and change, for instance, arising from volcanic activity, solar and orbital variations (Huybers and Curry, 2006; Miller et al.,

2012; McGregor et al., 2015; Abram et al., 2016; Gebbie and Huybers, 2019). In addition, there is the tendency of the climate system, broadly defined as the coupled ocean, atmosphere, land surface, land and sea ice, biota, and solid earth, to vary on a broad range of timescales under stochastic forcing (Hasselmann, 1976)."

Michael N. Evans Lucie J. Lücke Kevin J. Fan Feng Zhu

---

## Author Comment (AC3)

**Authors' Response:**

Thank you to all three reviewers for the insightful remarks and constructive criticisms of the submitted manuscript, database, and codebase. We are pleased to describe plans for revising the manuscript as follows (reviewer remarks in blue; our response in black.)

RC2: 'Comment on essd-2025-364', Julien Emile-Geay, 21 Aug 2025

Citation: <a href="https://doi.org/10.5194/essd-2025-364-RC2">https://doi.org/10.5194/essd-2025-364-RC2</a>

Review of "A database of databases for Common Era paleoclimate applications" by Evans et al

**Summary:**

The article presents an attempt at synthesizing paleoclimate proxy records across 5 different databases with partial overlap. A detailed procedure for identifying and removing duplicates is described, and two applications of analysis on the unified database are shown. This careful work will be suitable for publication after minor revisions.

**Scientific Comments:**

Given the scope of the journal, my comments will focus on the data and associated code.

1) Since this is, in part, an attempt at standardization, I would like to point out that many (3/5) of the constituent databases have recently been updated on lipdverse.org, and now use terminology that espouses the community-sourced LinkedEarth ontology (https://linked.earth/ontology/), which itself uses a number of controlled vocabularies (https://lipdverse.org/vocabulary/). Some of these vocabularies have been aligned to relevant terms in the NCEI PaST Thesaurus

(https://www.ncei.noaa.gov/products/paleoclimatology/paleoenvironmental-standard-terms-thesa urus). To the extent possible, it would be good to align the terms used in this study (cf Table 1) to those standards, and refer to them in the text so readers are more aware of them.

**Authors' response:**

We have mapped our compact common dictionary, which was originally based on the PAGES2k (2017) terminology, to match the first level of the LiPDverse vocabulary (<a href="https://lipdverse.org/vocabulary/">https://lipdverse.org/vocabulary/</a>) as follows:

- Terms which we have changed to match LiPDverse:
  - 'climateInterpretation variable' -> interpretation variable
  - 'climateInterpretation\_variableDetail' -> interpretation\_variableDetail
  - 'climateInterpretation\_direction' -> interpretation\_direction (new in v2.0)
  - 'climateInterpretation\_seasonality' -> interpretation\_seasonality (new in v2.0)

- 'paleoData\_variableName' (new in dod2k v2.0: see below, in response to Nick McKay's comments: name of the variable derived from the proxy observation, which may be different from the proxy observation)
- Terms which are already in agreement with the LiPDverse terminology:
  - archiveType, paleoData units
- Terms which have no clear match in the LiPDverse first level we have left the same.

However based on your suggestion we have also changed the terminology for the entries of the archiveType and paleoData\_proxy (e.g. we renamed 'tree' to 'Wood' and 'd2H' to 'dD' to comply with LiPDverse terminology).

We hope these changes will make the usage of the database and the code more convenient for users.

With regard to the versions of the curated databases, we will update the versions ingested into dod2k as follows:

- PAGES2k: we propose to use the lipd serialization version 2.2.0 that is on lipdverse.
- Iso2k: update from iso2k v1.0.1 to iso2k v1.1.2 using lipdverse current version.
- Breitenmoser et al (2014): no update is available.
- CoralHydro2k: update from v1.0.0 to v1.0.1 using lipdverse current version.
- Sisal: no update, as we are using v3 (2024), and lipdverse is at v2.1.1, although as of this writing the directories at <a href="https://lipdverse.org/SISAL-LiPD/current\_version/">https://lipdverse.org/SISAL-LiPD/current\_version/</a> are empty.
- 2) While the database itself will undeniably be useful for some applications, I believe the associated workflows are of greater value still. In particular, the workflow to identify and remove duplicates addresses a recurring issue in this line of work, and to my knowledge it is the first published instance of such a workflow being described in detail, and shared in code form.

Unfortunately, there is no universal standard for sharing workflows. It is very helpful that the authors made notebooks and auxiliary Python modules available through GitHub, but the notebooks are still a little rough around the edges (cf a lot of commented out old code) and lack a narrative. I would like to invite the authors to organize their cleaned-out notebooks as a JupyterBook, and share it through a gallery like PaleoBooks: https://linked.earth/PaleoBooks/. I believe the work will have greater visibility there, and will have more enduring value to the community.

Authors' response: We will clean up the notebooks and create a tutorial to bridge from the Quickstart Guide to the notebook workflow. We are using mkdocs (https://github.com/mkdocs/mkdocs) as mentioned in the response to reviewer 1 who made a similar request. We will discuss with linked.earth whether a PaleoBooks submission would be a useful contribution.

**Editorial Comments:**

The paper is well written, though I have a handful of suggestions.

there is inconsistent terminology throughout the manuscript, sometimes referring to PAGES2K, or PAGES2k. The proper nomenclature is PAGES 2k (lowercase, with space).

**Authors' response:**

We certainly can agree that a consistent nomenclature is good practice. The PAGES 2k Network conforms to the suggested nomenclature when referring to the research network. The reviewer's own lead authored data descriptor refers consistently to PAGES2k (no space), as do Iso2k and Coralhydro2k, when referring to databases themselves produced by that network, and we will do the same here.

L117: complimentary —> should be "complementary"

**Authors' response:**

We will make this revision. Thank you, here and elsewhere, for pointing out these errors.

Table 2: it looks like the authors loaded the constituent databases from various static files. For most up to date information on PAGES 2k, Iso2k, and CoralHydro2k, it is recommended that they download the latest from lipdverse.org, as many updates were made this summer.

**Authors' response:**

Thank you. We were not aware of updates to the lipdverse versions, having sourced from the NCEI and other repositories; we will make these revisions (see also notes earlier). We will modify the load notebooks to ask whether to download from source before loading from the local copy.

L167: "The evidently true duplicate records ...". How many such duplicates were found, and what fraction of the total number does that represent?

**Authors' response:**

This is given in the submitted manuscript, I. 198, first results sentence: "DoD2k v1 consists of 4516 records (4841 before duplicate screening)." We will update this result along with the planned updating of component databases.

L238: "the sensor model in PRYSM (Dee et al., 2015)," —> It should be noted that this is the sensor model introduced by Partin et al (2013), <a href="http://dx.doi.org/10.1130/G34718.1">http://dx.doi.org/10.1130/G34718.1</a>.

**Authors' response:**

Thank you, this is an important original citation to make, which we will add where indicated.

Section 4: it is odd to put results in the discussion. I recommend renaming Section 4 "Applications" and having a very short Section 5 called "discussion" or "outlook" that incorporates what is currently in Section 4.3. Conclusions could be merged in there too.

**Authors' response:**

We will create a new Section 5 from Section 4.3 to clearly separate Results from Discussion.

L310: references should be parenthetical (\citep{}, not \citet{}).

L327: "filtering by dictionary terms we have employed" —> filtering by THE dictionary terms we have employed (missing THE).

**Authors' response:**

We will make these revisions.

In surveying S\_analysis.ipynb, if appears that the authors re-invented the wheel in how they implemented something as straightforward as linear regression. I recommend that they use statsmodels (https://www.statsmodels.org/stable/), as it will provide all the information the authors need via model summaries, and lead to a more lightweight notebook, less prone to errors since statsmodels has been more thoroughly vetted.

**Authors' response:**

We will update the speleothem analysis notebook for use of the statsmodels package, and update the results in the manuscript accordingly.

**Julien Emile-Geay**

Michael N. Evans Lucie J. Lücke Kevin J. Fan Feng Zhu

---

## Author Comment (AC4)

**Authors' Response:**

Thank you to all three reviewers for the insightful remarks and constructive criticisms of the submitted manuscript, database, and codebase. We are pleased to describe plans for revising the manuscript as follows (reviewer remarks in blue; our response in black.)

RC3: 'Comment on essd-2025-364', Nicholas McKay, 06 Sep 2025

Citation: <a href="https://doi.org/10.5194/essd-2025-364-RC3">https://doi.org/10.5194/essd-2025-364-RC3</a>

In this manuscript, the authors describe the rationale and methodology behind the assembly of a "Database of Databases" for paleoclimate for Common Era, and include two use cases to demonstrate the potential utility of this merged data product. The manuscript is well written and illustrated and addresses a common problem: how to use a collection of related, but not custom-built, data compilations to address a problem that could benefit from a larger collection of data.

The authors identify metadata integration, including non-overlap and terminology differences, and duplicate handling as the primary challenges in this exercise. This is consistent with my experience. The approach and methodology for identifying, handling and tracking the choices made in the deduplication process is well done, and a valuable addition to the literature.

The code and data to load in the databases, align their terminology, and remove duplicates is all available, as is the code for the use cases and figures. As always, it's great to have access to all of this to get into the details of how the authors did what they did. I thank the authors for following best practice here! Using the instructions I was able to run all the notebooks except one ("load\_pages2k\_vv2.ipynb"), which seemed to hang during the `pdb = cfr.ProxyDatabase().fetch('PAGES2kv2')` command.

**Authors' response:**

We will modify the load\_pages2k notebook to load from the most recent lipdverse version, which is labeled pages2k v2.2.0 and has 647 lipid files. This includes the Palmyra (2013) update that was retrieved and used to replace the older Palmyra record, so this also serves to simplify the load\_pages2k database and also remove this problem for users (which we are unable to reproduce).

Having the codebase is very helpful, however I do think that it would be a challenge for others to try to build on this approach to add additional or alternate compilations using the same design. It's certainly more of a reference with examples than it is a tool to easily create new compilations.

Authors' response: We do not share the reviewer's certainty. We hope to convince them with the requested tutorial and improved commenting. We are encouraged by the other reviewers'

enthusiasm for the detection and decision notebooks, and we also point to the applications examples (with the caveats noted by this reviewer and discussed below).

Overall, I think this is a worthwhile contribution and I think the community will find the DoD very helpful. Indeed, I expect the database itself (as opposed to the methodologies described to create) to be widely used and a starting point for many researchers keen to take a data-intensive look at many aspects of Common Era climate. And because of this expectation, I have a few suggestions to make that database both more useful, and less prone to misuse.

First, while I appreciate the need for a reduced set of metadata while integrating across datasets, there are three additional fields that I think are critically needed. Fortunately, this metadata is available in most of the original compilations so the authors would not be starting from scratch.

1. interpretation direction: A field that describes whether the variable is positively or negatively related to the interpreted variable. The need for this field is evident in the first use case, where the authors multiply the tree ring and coral d18O datasets by -1 to allow for more direct comparison with the other proxies. Although it's true that these relationships due vary by proxy and archiveType, there are many examples of variable interpretations within archive and proxy classes. Lake sediment d18O is one example where this is often variable between lakes. Critically, there are also several examples in the literature where the interpretation direction was not properly applied, and this can lead to substantially wrong conclusions. Given its importance, including explicit interpretation direction for each interpreted dataset (and interpretation) is critical. After adding these metadata, I suggest replotting the PCA results using these metadata rather than the class-specific data.

**Author's response:**

This is a good idea, and we will implement this. But if we are not mistaken, this metadata field only exists for the PAGES 2k database and not for the other four databases: at most about  $692/4516 \sim 15\%$ .

We propose to keep this application subsection, but revise the discussion of the results to reflect the reviewer's concerns. We would note, as opposed to composite averaging, that a PCA on standardized proxy observations in their native variable simply identifies patterns of correlation, which could be either positive or negative in sign, between records. Our analysis did so by both archive and observation type in the largest subsets (Table 3).

We acknowledge that there might be differences in sign of linear regression coefficients across, for example, lake sediment d18O records, as the reviewer states; we would appreciate citations in support of this claim. And we acknowledge that the comparison of sign across the different PCAs shown is problematic, because the sign of each EOF pattern is arbitrary.

We also believe the results may disagree within and possibly across the possible comparison subsets, because there might not be a simple linear mapping or scaling from one environmental variable to the proxy observation, especially for observations which are influenced by both moisture and temperature in different ways and proportions.

Thus in agreement with the reviewer, we would expand on the uncertainties in doing so and interpreting the results, as we have just discussed. We originally wrote:

"However, perhaps because of differences in observational networks, time resolution and/or covariance estimation interval, there appears to be little agreement between PC1 and PC2 across archive and observation subsets. Although timeseries of the mean of PC1 show, at times, some agreement for certain archives (Figure 6 B), there is no agreement regarding PC2. All this suggests careful 280 additional analysis may be needed before a multi-archive, multi-observational analysis is performed and interpreted... Although there is some degree of regional spatial agreement of EOF sign within and across archives and observations, there are also many instances of disagreement of EOF sign within and across archives and observation types. Because spatial patterns may also be sensitive to observational network and the potential for both T and M influences in this subset, we again assess that more analysis within these archives and data types is needed before we can identify large scale patterns across the multi-archive and multi-observation database."

We would expand that discussion as described. We will include the reviewer's point that as as community, we may need to go back to the original publications to add metadata on the calibration and/or interpretation of individual records, enabling use of this information in such studies, before further analysis is warranted.

2. Seasonality: Many of the datasets are interpreted to be more sensitive to climate variability during some parts of the year than others, and it's very valuable to be able to filter by interpreted seasonality to test various hypotheses.

Authors' response: we agree this would be valuable, and we will add this. But again, the metadata have not been compiled for any of these datasets except PAGES2k, which is a small subset of the dod2k.

3. Variable name: The database includes both units (what units the variable was reported in) and proxy, which describes the type of physical, chemical, and/or biological systems that imprint climatic condition onto the archive, but not the name of the variable itself. This is often similar (or identical) to the proxy, but not always. For example, chironomid data are measured as count or assemblage data, but are included in the database as a calibrated temperature variable. So the proxy is correctly "chironomid" and the units are correctly "degC", but the variable name should be "temperature" or something similar. This is particularly useful for making coherent axes labels using the metadata form the database.

Authors' response: We see the reviewer's point - in addition to the example cited, there are for instance some tree-ring width temperature reconstructions that are included in the PAGES2k

database which we would ingest as tree-ring width data without this field. We propose to add an additional dictionary term, paleodata\_variableName, that can be defined as name of the variable derived from the proxy observation, which may be different from the proxy observation (see above), which is in paleodata\_proxy. This may help make the codebase and database metadata more useful for future users and for when these distinctions are important in future versions of component databases.

My final major suggestion is that, to the extent possible, citation information for the original studies is provided in the manuscript and in the database. These data compilations are critical for large-scale study of the Common Era, but also tend to make it more difficult for the authors of the original study to be credited. Users will likely be unable to cite all of the studies if they're using the whole databases, but many will only use a subset and it's helpful to enable citation of the original studies whenever possible. Ideally this would be an additional field in the database.

Authors' response: As reported in Table 1 of the manuscript, this information is already included in the dod2k in the field: originalDataURL: Original data URL URL/DOI for each record. For instance, the first record of the iso2k database has originalDataURL:

'http://doi.pangaea.de/10.1594/pangaea.676709'

which points us to the full reference for citation.

However, for the fe23 dataset, originalDataURL was pointing to each flat Arizona format .rwl file, which has only the contributor name, but not the full original reference. We will modify the fe23 load notebook to instead point to the NOAA template rwl file, which also includes the original reference information as supplied to NOAA/NCEI by the scientists who deposited the data in the repository.

We recognize the reviewer's request to also potentially list the more than 4,500 individual original sources in the reference list for the manuscript. We feel this would be unwieldy but also redundant to listing the originalDataURLs for each record, and to what is listed in each of the curated component databases and their published data descriptors.

Michael N. Evans Lucie J. Lücke Kevin J. Fan Feng Zhu